# Annotations Are Not All You Need: A Cross-modal Knowledge Transfer Network for Unsupervised Temporal Sentence Grounding

**Xiang Fang[1*]   Daizong Liu[2*]   Wanlong Fang[3*]   Pan Zhou[1†]   Yu Cheng[4]   Keke Tang[5]   Kai Zou[6]**

[1]Hubei Key Laboratory of Distributed System Security, Hubei Engineering Research Center on Big Data

Security, School of Cyber Science and Engineering, Huazhong University of Science and Technology   [2]Peking University

[3]Henan University   [4]The Chinese University of Hong Kong   [5]Guangzhou University   [6]Protagolabs Inc.

xfang9508@gmail.com    dzliu@stu.pku.edu.cn    wanlongfang@gmail.com

panzhou@hust.edu.cn    chengyu@cse.cuhk.edu.hk    tangbohutbh@gmail.com    kz@protagolabs.com

## Abstract

This paper addresses the task of temporal sentence grounding (TSG). Although many respectable works have made decent achievements in this important topic, they severely rely on massive expensive video-query paired annotations, which require a tremendous amount of human effort to collect in real-world applications. To this end, in this paper, we target a more practical but challenging TSG setting: unsupervised temporal sentence grounding, where both paired video-query and segment boundary annotations are unavailable during the network training. Considering that some other cross-modal tasks provide many easily available yet cheap labels, we tend to collect and transfer their simple cross-modal alignment knowledge into our complex scenarios: 1) We first explore the entity-aware object-guided appearance knowledge from the paired Image-Noun task, and adapt them into each independent video frame; 2) Then, we extract the event-aware action representation from the paired Video-Verb task, and further refine the action representation into more practical but complicated real-world cases by a newly proposed copy-paste approach; 3) By modulating and transferring both appearance and action knowledge into our challenging unsupervised task, our model can directly utilize this general knowledge to correlate videos and queries, and accurately retrieve the relevant segment without training. Extensive experiments on two challenging datasets (ActivityNet Captions and Charades-STA) show our effectiveness, outperforming existing unsupervised methods and even competitively beating supervised works.

## 1   Introduction

As a popular yet challenging natural language processing task, temporal sentence grounding (TSG)

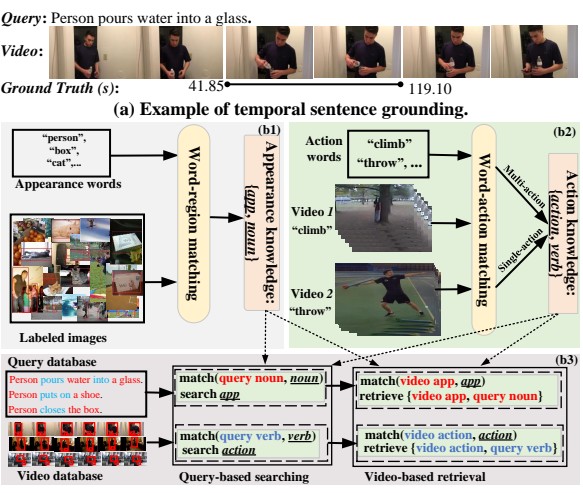

Figure 1: (a) Example of the temporal sentence grounding (TSG) task. (b) Illustration of our proposed CMKT network, where (b1) is the appearance knowledge learning module, (b2) is the action knowledge learning module, and (b3) is the knowledge transfer module, where "app" denotes "appearance". Note that there are no ground-truth annotations in (b3). Best viewed in color.

(Fang et al., 2022, 2023a; Liu et al., 2023c) has drawn increasing attention in recent years. TSG aims to locate a temporal video segment with an activity that semantically corresponds to a given sentence query. As shown in Figure 1(a), only a short video segment semantically matches the query, while most of the video contents are query-irrelevant. Clearly, TSG tries to break through the barrier between computer vision and natural language processing techniques for more challenging cross-modal grounding (Li et al., 2023b,a, 2022; Wang and Shi, 2023; Wang et al., 2021a, 2020c,a,b).

Most previous TSG works (Liu et al., 2023a; Fang et al., 2023b; Zhang et al., 2021; Fang et al., 2020, 2021b,a; Fang and Hu, 2020; Liu et al., 2021b, 2020b, 2022a, 2021a, 2022d, 2021c) are under a fully-supervised setting, where each frame is manually annotated as query-relevant or query-

---

*Equal contributions.   †Corresponding author.

irrelevant. Despite the decent progress, these data-hungry methods severely rely on numerous frame-query annotations, which are significantly labor-intensive and time-consuming to collect. To alleviate annotation reliance to a certain extent, some recent works (Tan et al., 2021; Chen et al., 2019; Mithun et al., 2019; Ma et al., 2020; Lin et al., 2020; Song et al., 2020; Zhang et al., 2020d) explore a weakly-supervised setting to only leverage the coarse-grained video-query annotations instead of the fine-grained frame-query annotations. Unfortunately, this weakly-supervised setting still requires expensive video-query annotations and its performance validates unsatisfactory.

To this end, in this paper, we try to tackle a more practical but challenging setting for the TSG task, *i.e.*, unsupervised TSG, which excludes both coarse-grained video-query annotations and fine-grained frame-query annotations. Therefore, how to guide the unsupervised model to extract the vision-language correlations becomes a crucial problem. Luckily, unlike the expensively annotated TSG task requiring long sentences and complicated videos, some cross-modal tasks (*e.g.*, image retrieval and video retrieval) always provide a massive number of cheap and fully-annotated datasets, *i.e.*, matched image-noun and paired video-verb. Hence, we pose a brand-new idea: *can we transfer the annotation knowledge from other cheap cross-modal annotations to the complicated and challenging unsupervised TSG task?* Our main motivation is that we can first learn simple image-noun and video-verb correlations to extract the visual appearance and action knowledge, respectively. Then, by generalizing such cross-modal knowledge and transferring it into unsupervised TSG, we can not only well correlate the queries with corresponding videos without large-scale video-query pairs, but also match them with image-aware frames without frame-query pairs. Therefore, the current problem becomes how to separately collect the appearance and action knowledge and adaptively transfer the knowledge into our unsupervised TSG task.

To tackle these issues, we propose a novel Cross-Modal Knowledge Transferring (CMKT) network, which fine-tunes and transfers the extracted appearance and action knowledge from other cross-modal tasks to search the best-matching video activity related to given queries in the unsupervised TSG setting. As shown in Figure 1(b), our CMKT contains three modules: (i) In the appearance knowledge module, we try to learn the object-guided matched region-noun information from the Image-Noun task to model the appearance information. (ii) In the action knowledge module, single-action and multi-action branches are designed to simulate complex video scenes. In the single-action branch, we extract the pure action information based on a set of labeled single-action videos. In the multi-action branch, since real-world videos often contain multiple actions, we introduce a clip-level copy-paste strategy to construct the action-hybrid videos to adapt to more complex scenes. (iii) Finally, we refine both the appearance and action knowledge, and aggregate and transfer them into our complicated unsupervised TSG for inference without further training.

Our main contributions are summarized as follows:

- To the best of our knowledge, we make the first attempt to transfer the cross-modal knowledge from other tasks into the TSG task. Without any TSG annotation, we can directly generalize the collected pre-trained appearance and action information for precise grounding, which eliminates complex training.

- We carefully design the appearance and action knowledge collection modules to learn simple knowledge from other cheap tasks. To handle the complicated video with multiple action contexts, we introduce a copy-paste idea to synthesize various multi-action clips, which improves the generalization ability.

- Comprehensive experiments on two challenging datasets (ActivityNet Captions and Charades-STA) show the effectiveness of our proposed CMKT.

## 2 Related Works

Most existing TSG methods are under the fully-supervised setting (Tang et al., 2021; Liu et al., 2023b; Zhang et al., 2021; Cao et al., 2020; Lei et al., 2020; Liu et al., 2020a, 2022b; Liu and Hu, 2022; Liu et al., 2022e), where all video-query pairs and precise segment boundaries are manually annotated. Based on the top-down approach, some methods (Anne Hendricks et al., 2017; Chen et al., 2018; Zhang et al., 2020b) first pre-define multiple segment proposals and then align these proposals with the query for cross-modal semantic matching based on the similarity. Finally, the

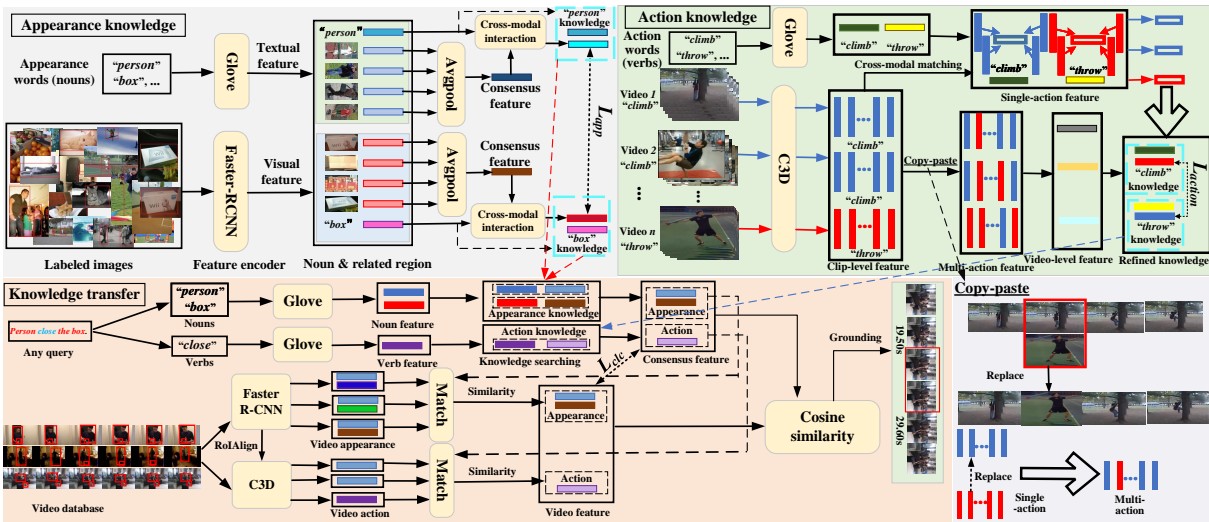

Figure 2: Overview of our CMKT framework for unsupervised TSG. We design two knowledge modules to extract both appearance and action knowledge from other simple yet cheap cross-modal tasks. (i) In appearance branch, we learn the region-word pair information and the appearance information can be viewed as the visual semantic of a given noun. (ii) The action knowledge module contains two branch: single-action branch and multi-action branch. For the single-action branch, we extract the pure action information based a set of labeled action video. Specifically, based on the branch, we can obtain the visual represent of the verbs. As for multi-action branch, since many video often contain multiple action, we introduce the clip-level copy-paste method to construct the action-hybrid videos. The global information in these videos contain the action features. (iii) Finally, we introduce the appearance and action knowledge into our TSG task by knowledge searching. Best viewed in color.

best-matched proposal is selected as the predicted segment. Recently, some works utilize the bottom-up strategy (Chen et al., 2020a; Mun et al., 2020; Zhang et al., 2020a), which directly regresses the boundary of the target segment or predicts boundary probabilities frame-wisely. To alleviate the reliance to a certain extent, some state-of-the-art turn to the weakly-supervised setting (Tan et al., 2021; Chen et al., 2019; Mithun et al., 2019; Ma et al., 2020; Lin et al., 2020; Song et al., 2020; Zhang et al., 2020d), where only video-query pairs are annotated without precise segment boundaries. To further mitigate the reliance, an unsupervised work (Liu et al., 2022c) clusters the semantic information of the whole query set by a complex training process for grounding. Different from the above methods, we address the challenging unsupervised TSG task from a novel perspective: collecting and transferring the general cross-modal knowledge from other simple yet cheap tasks.

## 3 Our Method

### 3.1 Overview

**Problem definition.** Given a set of untrimmed videos $V = \{V_h\}_{h=1}^{N^V}$ and sentence queries $Q = \{Q_g\}_{g=1}^{N^Q}$, we denote $V_h$ and $Q_g$ as the $h$-th video

and the $g$-th query, $N^V$ and $N^Q$ as the number of videos and queries, respectively. For each query, unsupervised TSG has no annotation information about its corresponding related video and detailed activity location. Obviously, our unsupervised TSG is more challenging than previous supervised TSG, since we drop all annotated information between $V$ and $Q$ including their correspondence and annotated segment boundaries.

**Overall pipeline.** To tackle the challenging unsupervised setting, we propose a novel Cross-Modal Knowledge Transfer (CMKT) network shown in Figure 2, which first designs two task-agnostic knowledge collection modules to learn both appearance and action knowledge from other labeled but cheap cross-modal tasks, then refine and transfer their knowledge into our complicated TSG task.

For the appearance knowledge module, we first collect a set of conceptual nouns and their corresponding image regions from public Image-Noun datasets (*e.g.*, Visual Genome (VG) (Krishna et al., 2017)), where a noun corresponds to multiple relevant regions. We denote these region-level images as $V^a = \{\boldsymbol{v}_i^a\}_{i=1}^{N^a}$ and their corresponding words (object nouns) as $W^n = \{\boldsymbol{w}_j^n\}_{j=1}^{N^n}$, where $N^a$ and $N^n$ are the number of regions and nouns, respectively. Then, we explore the region-level

appearance knowledge by the region-word pair information $(\boldsymbol{w}_j^n, \bar{\boldsymbol{v}}_j^a)$, where $\bar{\boldsymbol{v}}_j^a$ is the consensus of all the regions corresponding to $\boldsymbol{w}_j^n$. For noun $\boldsymbol{w}_j^n$, we compute $\bar{\boldsymbol{v}}_j^a$ by averaging the representations of all relevant regions. With pairwise nouns and their consensus region representations, we can obtain general cross-modal appearance knowledge.

In the action knowledge module, we are given a massive number of conceptual verbs and their corresponding action-pure clips from popular Video-Verb datasets (*e.g.*, Kinetics (Carreira and Zisserman, 2017)), where a verb corresponds to many labeled clips. These clips and corresponding verbs are denoted as $V^m = \{\boldsymbol{v}_i^m\}_{i=1}^{N^m}$ and $W^v = \{\boldsymbol{w}_j^v\}_{j=1}^{N^v}$ respectively, and $N^m$ and $N^v$ are the number of clips and verbs. We tend to learn the clip-level action knowledge $(\boldsymbol{w}_j^v, \bar{\boldsymbol{v}}_j^m)$, where $\bar{\boldsymbol{v}}_j^m$ is the consensus of all the regions corresponding to $\boldsymbol{w}_j^v$. Besides obtaining the basic action feature via averaging over all relevant single-action clips, we synthesize the multi-action clips by the copy-paste approach (Dvornik et al., 2018; Xu et al., 2021; Rong et al., 2021) for simulating more complicated scenes. By refining both single- and multi-action representations, we obtain the final action knowledge. Finally, a knowledge transfer module is designed to transfer appearance and action knowledge into our unsupervised TSG.

## 3.2 Appearance Knowledge Collection

Since the cheap Image-Word datasets provide sufficient known image-noun pairs, we can leverage them to extract the simple cross-modal appearance knowledge. However, our desired appearance knowledge has two key properties: 1) Region-aware: Rather than extract visual backgrounds, appearance knowledge is required to accurately align the region-level foreground objects and corresponding nouns. It is more fine-grained than other methods that roughly link the whole image to words. 2) One-to-one matching: Instead of aligning a noun to multiple related images in a one-to-many manner, we focus on cross-modal one-to-one matching to alleviate the appearance variations of objects by aligning the noun and its corresponding consensus region. Therefore, we collect all the nouns and their related regions on the VG dataset, which contains diverse contents that are useful to improve knowledge generalization.

**Noun-aware textual encoder.** Given $N^n$ nouns as the label set of $N^a$ region-level images, we utilize

Glove (Pennington et al., 2014) to embed each noun into a dense vector $\{\boldsymbol{w}_j^n\}_{j=1}^{N^n}$, where $\boldsymbol{w}_j^n \in \mathbb{R}^d$ is the $j$-th noun feature and $d$ is the feature dimension.
**Appearance-aware visual encoder.** Given $N^a$ region-level images, we use Faster-RCNN (Ren et al., 2015) to extract their features $\{\boldsymbol{v}_i^a\}_{i=1}^{N^a}$, where $\boldsymbol{v}_i^a \in \mathbb{R}^d$ is the $i$-th appearance feature.
**Appearance knowledge acquisition.** For the $j$-th noun $w_j^n$, we can obtain its prototypical region representation $\bar{\boldsymbol{v}}_j^a$ by averaging all relevant region features $\{\boldsymbol{v}_{j,k}^a\}_{k=1}^{K_j^n}$:

$$\bar{\boldsymbol{v}}_j^a = AVP(\boldsymbol{v}_{j,1}^a, \cdots, \boldsymbol{v}_{j,K_j^n}^a), \qquad (1)$$

where $\bar{\boldsymbol{v}}_j^a$ is the consensus appearance feature and $AVP(\cdot)$ denotes the average pooling, $K_j^n$ is the total related region number. To ensure that $\bar{\boldsymbol{v}}_j^a$ shares the same semantics with its corresponding noun-aware features, we learn two parametric transformation matrices $\boldsymbol{B}^a$ and $\boldsymbol{B}^n$ via a regularization loss (Lin et al., 2017) $\mathcal{L}_{app}$ to enforce consensus region feature $\boldsymbol{v}_j^a$ as close as possible to corresponding noun feature $\boldsymbol{w}_j^n$:

$$\mathcal{L}_{app} = ||\boldsymbol{B}^a \overline{\boldsymbol{V}}^a - \boldsymbol{B}^n \boldsymbol{W}^n||_F^2, \qquad (2)$$

where $||\cdot||_F$ is Frobenius norm, $\overline{\boldsymbol{V}}^a = \{\bar{\boldsymbol{v}}_j^a\}_{j=1}^{N^n} \in \mathbb{R}^{d \times N^n}$ and $\boldsymbol{W}^n = \{\boldsymbol{w}_j^n\}_{j=1}^{N^n} \in \mathbb{R}^{d \times N^n}$.

## 3.3 Action Knowledge Collection

Similarly, we can utilize cheap Video-Verb datasets to extract cross-modal action knowledge by learning the video-verb pairs. To meet the requirement of diverse actions in the TSG task, we collect the verbs and their paired trimmed videos from a highly diverse video dataset Kinetics (Carreira and Zisserman, 2017). In the TSG task, it is difficult to obtain the specific feature of every single action from multi-action videos. Thus, we simulate various action videos to better generalize the mixed action features. To collect different types of action knowledge, we design two branches: single- and multi-action branches. In the single-action branch, since all the clips share the same-class action information, we aim to obtain the pure average action feature of a given verb. For the multi-action branch, we refine the consensus action feature by merging multiple action clips into a hybrid one with a copy-paste strategy.
**Verb-aware textual encoder.** Given $N^v$ verbs, we use the Glove network (Pennington et al., 2014) to embed each verb into a dense vector $\boldsymbol{W}^v = \{\boldsymbol{w}_j^v\}_{j=1}^{N^v} \in \mathbb{R}^{d \times N^v}$.

**Action-aware visual encoder.** Given $N^m$ labeled trimmed videos, we first extract their clipwise features by a pre-trained C3D network (Tran et al., 2015), and then employ a multi-head self-attention (Vaswani et al., 2017) module to capture the long-range dependencies among video frames. We denote the extracted video features as $\boldsymbol{V}^m = \{\boldsymbol{v}_i^m\}_{i=1}^{N^m} \in \mathbb{R}^{d \times N^m}$.

**Action knowledge acquisition.** In the *single-action* branch, we are given a set of verbs $\{\boldsymbol{w}_j^v\}_{j=1}^{N^v}$ and each verb $\boldsymbol{w}_j^v$ corresponds to multiple action-pure videos $\{\boldsymbol{v}_{j,p}^m\}_{p=1}^{N_j^m}$, where $N_j^m$ is the number of action videos corresponding to $\boldsymbol{w}_j^v$. For verb $\boldsymbol{w}_j^v$, we directly utilize the average pooling to obtain its average action feature: $(\boldsymbol{v}_j^m)' = AVP(\boldsymbol{v}_{j,1}^m, \ldots, \boldsymbol{v}_{j,N_j^m}^m)$.

Since real-world videos often contains multiple actions, in the *multi-action* branch, we introduce a copy-paste strategy to augment the multi-action videos. As shown in Figure 2, given any two trimmed action videos, we randomly copy some clips from a video, and paste them onto the other video by replacing the same number of clips. For action-pure video $\boldsymbol{v}_{j,p}^m$ with $N_{j,p}^m$ clips (corresponding to verb $\boldsymbol{w}_j^v$) and another action-pure video $\boldsymbol{v}_{x,z}^m$ (irrelevant to verb $\boldsymbol{w}_j^v$), we randomly replace $P_{j,p}^m$ clips of $\boldsymbol{v}_{j,p}^m$ with $P_{j,p}^m$ clips from $\boldsymbol{v}_{x,z}^m$. The augmented multi-action video is denoted as $(\boldsymbol{v}_{j,p}^m)^*$. For convenience, we design a metric to denote the copy-paste amount: paste rate $\mu = P_{j,p}^m/N_{j,p}^v \in (0,1)$, *i.e.*, the percent of replaced clips in the original video $\boldsymbol{v}_{j,p}^m$. After the copy-paste process, since the augmented video $(\boldsymbol{v}_{j,p}^m)^*$ still contains verb-related action information (related to verb $\boldsymbol{w}_j^v$), we obtain the corresponding average action feature by the average pooling: $(\boldsymbol{v}_j^m)'' = AVP((\boldsymbol{v}_{j,1}^m)^*, \ldots, (\boldsymbol{v}_{j,N_j^m}^m)^*)$. $(\boldsymbol{v}_j^m)''$ denotes the generated action feature of $\boldsymbol{v}_j^m$ after the copy-paste operation. By combining single-action $(\boldsymbol{v}_j^m)'$ and multi-action $(\boldsymbol{v}_j^m)''$, the final consensus action feature is obtained by:

$$\bar{\boldsymbol{v}}_j^m = (1+\mu)(\boldsymbol{v}_j^m)' + (1-\mu)(\boldsymbol{v}_j^m)'', \qquad (3)$$

where $\mu_c$ is used to adaptively control the balance between the action-pure information and hybrid action information.

To generalize the above action knowledge, we design an *inter-action* loss $\mathcal{L}_{inter}$ and an *intra-action* loss $\mathcal{L}_{intra}$ for constraint. For the *inter-action* loss, in different augmented videos, we take the matched action-verb pairs as positive samples and the unmatched action-verb pairs as negative samples, and use the weighted binary cross-entropy loss to supervise the verb-relevance $\{\boldsymbol{r}_j^{wv}\}_{j=1}^{N^v}$:

$$\mathcal{L}_{inter} = \sum\nolimits_{j=1}^{N^v}(-\boldsymbol{y}_j \log(\boldsymbol{r}_j^{wv}) - (1-\boldsymbol{y}_j)\log(1-\boldsymbol{r}_j^{wv})),$$

where $\boldsymbol{y}_j$ the label of the $j$-th sample that equals 1 for the matched action-verb pairs and 0 for the unmatched action-verb pairs; $\boldsymbol{r}_j^{wv}$ is obtained by

$$\boldsymbol{r}_j^{wv} = \frac{\bar{\boldsymbol{v}}_j^m(\boldsymbol{w}_j^v)^T}{||\bar{\boldsymbol{v}}_j^m||_2^2 ||\boldsymbol{w}_j^v||_2^2}. \qquad (4)$$

For the *intra-action* loss, in each multi-action video, we take verb-related video clips as positive samples and define verb-irrelevant video clips as negative samples. We adopt a hinge loss for supervision:

$$\mathcal{L}_{intra} = \sum\nolimits_{j=1}^{N^v} max(0, \beta - \boldsymbol{r}_j^{wv} + \hat{\boldsymbol{r}}_j^{wv}),$$

where $\beta$ is a parameter and $\hat{\boldsymbol{r}}_j^{wv}$ denotes the verb-relevance of negative clips. Thus, the final action knowledge loss is:

$$\mathcal{L}_{action} = \mathcal{L}_{inter} + \mathcal{L}_{intra}. \qquad (5)$$

Eq. (5) tries to align the same-class action features and push the diff-class action features away.

### 3.4 Knowledge Transfer to Unsupervised TSG

In previous sections, we can obtain consensus appearance knowledge and consensus action knowledge, which are important for comprehending the target moments in our TSG task. However, we cannot directly utilize the collected knowledge in our TSG task due to (i) the domain gap between Image-Noun/Video-Verb task and the TSG task, and (ii) more complicated scenes in the TSG task. Thus, we aim to fine-tune collected knowledge to fit the TSG domain by knowledge transfer. In the following, we will illustrate how we transfer the collected knowledge into the TSG task.

**TSG query encoder.** Given a set of queries $Q = \{Q_g\}_{g=1}^{N^Q}$, to keep the query features same as the knowledge-based one, we utilize the Glove network to embed each word into a dense vector. For $Q_g$ with $N_g^q$ words, its word-level features are denoted as $Q_g = \{\boldsymbol{f}_{g,j}^q\}_{j=1}^{N_g^q} \in \mathbb{R}^{d \times N_g^q}$. We utilize the NLP tool spaCy (Honnibal and Montani, 2017) to parse nouns from the given query, then remove duplicate nouns. The textual features of the reserved nouns are denoted as $\{\boldsymbol{f}_{g,j}^n\}_{j=1}^{N_g^n}$, where $\boldsymbol{f}_{g,j}^n \in \mathbb{R}^d$ is the $j$-th noun feature and $N_g^n$ is the reserved noun number. Similarly, we parse verbs and obtain the reserved verb feature set $\{\boldsymbol{f}_{g,j}^v\}_{j=1}^{N_g^v}$, where $\boldsymbol{f}_{g,j}^v \in \mathbb{R}^d$ is the $j$-th reserved verb feature and $N_g^v$ is the reserved verb number.

**TSG video encoder.** Given $N^V$ videos $V = \{V_h\}_{h=1}^{N^V}$, we denote $V_h$ as the $h$-th video and $T_h$ as its frame number. For the appearance feature,

we utilize Faster-RCNN (Ren et al., 2015) built on a ResNet50 (He et al., 2016) backbone to detect $K$ objects on each video to obtain a set of appearance features $\boldsymbol{V}_h^a = \{\boldsymbol{f}_{h,i,k}^a\}_{i=1,k=1}^{i=T_h,k=K}$. For the action feature, we first divide video $V_h$ into $T_h/8$ clips. Then, we utilize a pre-trained C3D network (Tran et al., 2015) and a multi-head self-attention to extract the clip-aware features $\boldsymbol{V}_h^m = \{\boldsymbol{f}_{h,i}^m\}_{i=1}^{T_h/8}$.

**Fitting the TSG domain via knowledge transfer.** To transfer the pre-trained knowledge into the TSG task, we fine-tune the collected appearance and action knowledge based on the principle of cross-modal cycle consistency. Given the nouns features $\{\boldsymbol{f}_{g,j}^n\}_{j=1}^{N_g^n}$ and verbs features $\{\boldsymbol{f}_{g,j}^v\}_{j=1}^{N_g^v}$, we search each $\boldsymbol{f}_{g,j}^n$ in the appearance knowledge and each $\boldsymbol{f}_{g,j}^v$ in the action knowledge with the cosine similarity, then re-represent them with the collected consensus appearance features $\overline{\boldsymbol{v}}_j^a$ and action features $\overline{\boldsymbol{v}}_j^m$, respectively. For convenience, we denote $\boldsymbol{A}_g = \{\overline{\boldsymbol{v}}_j^a\}_{j=1}^{N_g^n} \in \mathbb{R}^{d \times N_g^n}$ and $\boldsymbol{M}_g = \{\overline{\boldsymbol{v}}_j^m\}_{j=1}^{N_g^v} \in \mathbb{R}^{d \times N_g^v}$. As for the visual appearance and action features, we also enhance them via the collected knowledge, respectively.

To fine-tune consensus features ($\boldsymbol{A}_g$ and $\boldsymbol{M}_g$), we design a cross-modal cycle-consistent loss to make collected knowledge more applicable to the TSG task. For appearance knowledge, we first compute the cosine similarity matrix $\boldsymbol{S}_g^n \in \mathbb{R}^{N_g^n \times N^V}$ between noun-guided consensus appearance features and video appearance features. Then, we combine all appearance features by treating similarities as weights to obtain reconstructed nouns features:

$$\widetilde{\boldsymbol{Q}}_g^n = \boldsymbol{W}_g^n \boldsymbol{C}^a \sigma_c(\boldsymbol{S}_g^n)^T,$$
$$\boldsymbol{S}_g^n = (\boldsymbol{W}_g^n \boldsymbol{A}_g)^T \boldsymbol{W}_g^n \boldsymbol{C}^a, \qquad (6)$$

where $\boldsymbol{C}^a = \{\boldsymbol{V}_h^a\}_{h=1}^{N^V}$, $\boldsymbol{W}_g^n$ is a learnable transformation matrix, $\sigma_c(\cdot)$ is the softmax operation along the column dimension, and $\widetilde{\boldsymbol{Q}}_g^n$ is the reconstructed noun representation. Similarly, we combine all action features to obtain the reconstructed verbs features:

$$\widetilde{\boldsymbol{Q}}_g^v = \boldsymbol{W}_g^v \boldsymbol{C}^m \sigma_c(\boldsymbol{S}_g^v)^T,$$
$$\boldsymbol{S}_g^v = (\boldsymbol{W}_g^v \boldsymbol{M}_g)^T \boldsymbol{W}_g^v \boldsymbol{C}^m, \qquad (7)$$

where $\boldsymbol{C}^m = \{\boldsymbol{V}_h^m\}_{h=1}^{N^V}$, $\boldsymbol{W}_g^v$ is a learnable transformation matrix, $\widetilde{\boldsymbol{Q}}_g^v$ is the reconstructed verb features, and $\boldsymbol{S}_g^v \in \mathbb{R}^{N_g^v \times N^V}$ is the similarity matrix between transformed verb and action features.

Our cycle-consistent loss aims to (i) align the semantics of original nouns and their reconstructed ones, (ii) pull verbs and their reconstructed ones together. Thus, we construct two indicator matrices

$\widetilde{\boldsymbol{Y}}_g^a \in \mathbb{R}^{N_g^n \times N_g^n}$ and $\widetilde{\boldsymbol{Y}}_g^m \in \mathbb{R}^{N_g^v \times N_g^v}$ to evaluate if each noun/verb and its reconstructed one are the same or not as:

$$\widetilde{\boldsymbol{Y}}_g^a = \sigma((\widetilde{\boldsymbol{Q}}_g^n)^T(\boldsymbol{W}_g^n \boldsymbol{A}))^T,$$
$$\widetilde{\boldsymbol{Y}}_g^m = \sigma((\widetilde{\boldsymbol{Q}}_g^v)^T(\boldsymbol{W}_g^v \boldsymbol{M}))^T. \qquad (8)$$

With $\widetilde{\boldsymbol{Y}}_g^a$ and $\widetilde{\boldsymbol{Y}}_g^m$, we optimize $\boldsymbol{W}_g^n$ and $\boldsymbol{W}_g^v$ by minimizing the cross-entropy loss:

$$\mathcal{L}_{clc} = -\sum_{g=1}^{N^Q}(\sum_{z=1}^{N_g^n}(\boldsymbol{y}_{g,z}^a)^T \log(\widetilde{\boldsymbol{Y}}_{g,z}^a)$$
$$+ \sum_{x=1}^{N_g^v}(\boldsymbol{y}_{g,x}^m)^T \log(\widetilde{\boldsymbol{Y}}_{g,x}^m)),$$

$\boldsymbol{y}_{g,z}^a$ and $\boldsymbol{y}_{g,x}^m$ are one-hot ground-truth vectors, where the $z$-th and $x$-th value are one, and rest values are zeros. By $\mathcal{L}_{clc}$, we use $\boldsymbol{W}_g^n$ and $\boldsymbol{W}_g^v$ to transfer all consensus appearance and action knowledge into TSG.

## 3.5 Knowledge-based Grounding

By pre-training the appearance knowledge by $\mathcal{L}_{app}$ and the action knowledge by $\mathcal{L}_{action}$ with the knowledge transfer by $\mathcal{L}_{clc}$, we can directly utilize the suitable transferred appearance knowledge $(\boldsymbol{w}_j^n, \overline{\boldsymbol{v}}_j^a)$ and action knowledge $(\boldsymbol{w}_j^n, \overline{\boldsymbol{v}}_j^a)$ to localize the target segment boundary *without any further training*. Specifically, during the inference, given some videos $\{V_h\}_{h=1}^{N^V}$ and a query set $\{Q_g\}_{g=1}^{N^Q}$, we first extract their appearance feature $\boldsymbol{f}_{h,i,k}^a$, action feature $\boldsymbol{f}_{h,i}^m$, noun feature $\boldsymbol{f}_{g,j}^n$, verb feature $\boldsymbol{f}_{g,j}^v$, respectively. Then, based on the collected appearance knowledge and action knowledge modules, we can obtain the corresponding consensus appearance and action features. We compute the cosine similarity matrix between consensus appearance features and video appearance features, and also calculate the cosine similarity matrix between consensus action features and video action features. After that, following (Liu et al., 2022c), we select the best-matched clip with the highest scores (sum of appearance similarity and action similarity) in the target video as the target segment center. We add the left/right frames into the segment if the ratio of their scores to the frame score of the closest segment boundary is less than a threshold $\gamma$. We repeat this step until no frame can be added. In this way, we can generate the final segment boundary.

Table 1: Performance comparison with state-of-the-art methods. 'Type' refers to supervision level, FS: fully-supervised setting, WS: weakly-supervised setting, US: unsupervised setting.

| Method | Type | ActivityNet Captions | | | | | Charades-STA | | | |
|---|---|---|---|---|---|---|---|---|---|---|
| | | IoU=0.1 | IoU=0.3 | IoU=0.5 | IoU=0.7 | mIoU | IoU=0.3 | IoU=0.5 | IoU=0.7 | mIoU |
| CTRL (Gao et al., 2017) | FS | 49.10 | 28.70 | 14.00 | - | 20.54 | - | 21.42 | 7.15 | - |
| 2D-TAN (Zhang et al., 2020b) | FS | - | 58.75 | 44.05 | 27.38 | - | - | 42.80 | 23.25 | - |
| LGI (Mun et al., 2020) | FS | - | 58.52 | 41.51 | 23.07 | 41.13 | 72.96 | 59.46 | 35.48 | 51.38 |
| VSLNet (Zhang et al., 2020a) | FS | - | 63.16 | 43.22 | 26.16 | 43.19 | 70.46 | 54.19 | 35.22 | 50.02 |
| VCA (Wang et al., 2021c) | WS | 67.96 | 50.45 | 31.00 | - | 33.15 | 58.58 | 38.13 | 19.57 | 38.49 |
| RTBPN (Zhang et al., 2020c) | WS | 73.73 | 49.77 | 29.63 | - | - | 60.04 | 32.36 | 13.24 | - |
| CTF (Chen et al., 2020b) | WS | 74.20 | 44.30 | 23.60 | - | 32.20 | 39.80 | 27.30 | 12.90 | 27.30 |
| MARN (Song et al., 2020) | WS | - | 47.01 | 29.95 | - | - | 48.55 | 31.94 | 14.81 | - |
| SCN (Lin et al., 2020) | WS | 74.48 | 47.23 | 29.22 | - | - | 42.96 | 23.58 | 9.97 | - |
| BAR (Wu et al., 2020) | WS | - | 49.03 | 30.73 | - | - | 44.97 | 27.04 | 12.23 | - |
| CCL (Zhang et al., 2020d) | WS | - | 50.12 | 31.07 | - | - | - | 33.21 | 15.68 | - |
| LCNet (Yang et al., 2021) | WS | 78.58 | 48.49 | 26.33 | - | 34.29 | 59.60 | 39.19 | 18.87 | 38.94 |
| CNM (Zheng et al., 2022) | WS | 78.13 | 55.68 | 33.33 | - | - | 60.39 | 35.43 | 15.45 | - |
| WSTAN (Wang et al., 2021b) | WS | 79.78 | 52.45 | 30.01 | - | - | 43.39 | 29.35 | 12.28 | - |
| DSCNet (Liu et al., 2022c) | US | - | 47.29 | 28.16 | - | - | 44.15 | 28.73 | 14.67 | - |
| **Our CMKT** | US | **73.35** | **50.69** | **31.28** | **16.42** | **38.79** | **47.80** | **30.96** | **18.87** | **30.42** |

# 4 Experiment

## 4.1 Datasets and Evaluation Metrics

**ActivityNet Captions.** From ActivityNet v1.3 (Heilbron et al., 2015; Lan et al., 2022), ActivityNet Captions contains 20,000 YouTube videos and 100,000 language queries. On average, a video is 2 minutes and a query has about 13.5 words. Following the public split (Gao et al., 2017), we utilize 17,031 video-query pairs for testing.

**Charades-STA.** Built upon the Charades dataset (Sigurdsson et al., 2016), Charades-STA contains 16,128 video-sentence pairs. The average video length is 0.5 minutes. The language annotations are generated by sentence decomposition and keyword matching with manual checks. Following (Gao et al., 2017), we remove 12,408 training pairs and utilize the others for testing.

**Evaluation metrics.** Following previous works (Mun et al., 2020; Zhang et al., 2020a), we use "IoU=m" for evaluation, which denotes the percentage of queries having at least one result whose Intersection over Union (IoU) with ground truth is larger than $m$. In our experiments, $m \in \{0.3, 0.5, 0.7\}$ for Charades-STA and $m \in \{0.1, 0.3, 0.5, 0.7\}$ for ActivityNet Captions. Also, we utilize mean IoU (mIoU) as the averaged temporal IoU between the predicted boundary and the ground-truth one.

## 4.2 Implementation details

For the appearance knowledge module, we utilize all the words on the Visual Genome dataset, so the noun number of region-level semantics is 27,801. The feature dimension $d$ is set to 512 for all the features. For the action knowledge module, we use the pre-trained C3D (Tran et al., 2015) network to extract the single-action and multi-action information on the Kinetics dataset. Also, we use all the verbs on the Kinetics dataset. To make a fair comparison with previous works (Gao et al., 2017; Zhang et al., 2020b), we also utilize the pre-trained C3D model to extract video features and employ the Glove model to obtain word embeddings. As some videos are too long, we set the length of video feature sequences to 128 for Charades-STA and 256 for ActivityNet Captions, respectively. We fix the query length to 10 in Charades-STA and 20 in ActivityNet Captions. Threshold $\gamma$ is set to 0.9 on Charades-STA and 0.8 on ActivityNet Captions. All the experiments are implemented by PyTorch.

## 4.3 Comparison with State-of-the-Arts

We conduct performance comparison with three categories of state-of-the-art TSG methods: (i) Fully-supervised (FS) setting (Gao et al., 2017; Zhang et al., 2020b; Mun et al., 2020; Zhang et al., 2020a); (ii) Weakly-supervised (WS) setting (Chen et al., 2020b; Lin et al., 2020; Wang et al., 2021b; Wu et al., 2020; Song et al., 2020; Zhang et al., 2020d; Huang et al., 2021; Wang et al., 2021c; Yang et al., 2021; Zhang et al., 2020c; Zheng et al., 2022); (iii) Unsupervised (US) setting (Liu et al., 2022c).

Table 1 reports the performance comparison. Without any supervision, our CMKT still obtains the competitive performance to WS and US methods on the ActivityNet Captions dataset. Compared with the US method DSCNet, our CMKT improves the performance by 3.40% and 3.12% in

Table 2: Main ablation study, where we remove each key component to investigate its effectiveness. "AK" denotes "appearance knowledge", "MK" denotes "action knowledge", "KT" denotes "knowledge transfer".

| Model | AK | MK | KT | ActivityNet Captions | | | | Charades-STA | | | |
|---|---|---|---|---|---|---|---|---|---|---|---|
| | | | | IoU=0.3 | IoU=0.5 | IoU=0.7 | mIoU | IoU=0.3 | IoU=0.5 | IoU=0.7 | mIoU |
| CMKT(a) | ✔ | ✔ | ✗ | 45.81 | 24.79 | 11.52 | 31.36 | 40.81 | 24.72 | 10.88 | 25.24 |
| CMKT(b) | ✗ | ✔ | ✔ | 47.50 | 26.82 | 12.37 | 32.71 | 42.43 | 26.87 | 11.74 | 26.90 |
| CMKT(c) | ✔ | ✗ | ✔ | 48.02 | 27.91 | 13.96 | 34.05 | 44.35 | 27.19 | 12.62 | 27.86 |
| CMKT(Full) | ✔ | ✔ | ✔ | **50.69** | **31.28** | **16.42** | **38.79** | **47.80** | **30.96** | **18.87** | **30.42** |

Table 3: Necessity of average pooling on ActivityNet Captions, where "MIP" means min-pooling, "MAP" means max-pooling, and "AVP" means average pooling.

| Module | Changes | IoU=0.3 | IoU=0.5 | IoU=0.7 | mIoU |
|---|---|---|---|---|---|
| Appearance knowledge | MIP | 47.80 | 29.08 | 13.76 | 35.70 |
| | MAP | 48.03 | 30.72 | 14.45 | 36.68 |
| | AVP | **50.69** | **31.28** | **16.42** | **38.79** |
| Action knowledge | MAP | 48.36 | 28.72 | 15.45 | 35.53 |
| | MIP | 49.27 | 29.26 | 16.20 | 36.61 |
| | AVP | **50.69** | **31.28** | **16.42** | **38.79** |

Table 4: Effect of action knowledge on ActivityNet Captions.

| Single-action | Multi-action | IoU=0.3 | IoU=0.5 | IoU=0.7 | mIoU |
|---|---|---|---|---|---|
| ✗ | ✔ | 48.21 | 27.70 | 15.55 | 36.03 |
| ✔ | ✗ | 49.36 | 29.82 | 15.73 | 37.54 |
| ✔ | ✔ | **50.69** | **31.28** | **16.42** | **38.79** |

Table 5: Performance of different knowledge transfer on ActivityNet Captions.

| Appearance | Action | IoU=0.3 | IoU=0.5 | IoU=0.7 | mIoU |
|---|---|---|---|---|---|
| ✗ | ✗ | 45.81 | 24.79 | 11.52 | 31.36 |
| ✗ | ✔ | 46.07 | 28.30 | 13.15 | 35.08 |
| ✔ | ✗ | 47.15 | 29.98 | 12.74 | 34.86 |
| ✔ | ✔ | **50.69** | **31.28** | **16.42** | **38.79** |

Table 6: Effect of copy-paste strategy on ActivityNet Captions.

| Changes | IoU=0.3 | IoU=0.5 | IoU=0.7 | mIoU |
|---|---|---|---|---|
| w/o copy-paste | 45.98 | 26.94 | 11.02 | 34.75 |
| w/ video connection | 49.22 | 30.76 | 14.54 | 37.82 |
| w/ copy-paste | **50.69** | **31.28** | **16.42** | **38.79** |

terms of "IoU=0.3" and "IoU=0.5", demonstrating the effectiveness of our proposed CMKT. On the Charades-STA dataset, compared with supervised methods (FS and WS), our CMKT in the US setting reaches competitive performance. Although our CMKT eliminates the training process, it outperforms US method DSCNet by 3.65%, 2.23% and 4.20% in terms of "IoU=0.3", "IoU=0.5" and "IoU=0.7", which shows our effectiveness.

### 4.4 Ablation study

**Main ablation studies.** To examine the effectiveness of each module, we conduct the main ablation studies in Table 2. CMKT(a) is the baseline model, which only utilizes pre-trained appearance and action features for inference without fine-tuning knowledge. CMKT(b) removes appearance knowledge module, and directly use the action knowledge to locate the target segment. In CMKT(c), we only use the appearance information to obtain the segment boundary by removing the action information. CMKT(Full) is our full CMKT model. CMKT(Full) performs better than all the ablation models, demonstrating the effectiveness of appearance and action knowledge for understanding the long yet complex sentence in TSG.

**Investigation on the consensus features.** To assess the effectiveness of average pooling (AVP) during the generation process of consensus appearance features, we compare different pooling strategies. As shown in Table 3, our AVP significantly outperforms both min-pooling (MIP) and max-pooling

(MAP) on both two knowledge collection modules, showing that AVP is suitable for integrating the information among a single modality for representing the same semantic as the other modality.

**Influence on different types of action knowledge.** To investigate the contribution of single- and multi-action videos to our final consensus action features, we conduct the corresponding ablations in Table 4 to evaluate the significance of single-action and multi-action knowledge. Obviously, both single-action and multi-action features contribute to our action knowledge module.

**Investigation on different types of knowledge transfer.** To evaluate the importance of appearance and action knowledge, we conduct corresponding experiments. As shown in Table 5, each knowledge contributes a lot to transferring the knowledge into our task. By jointing appearance and action knowledge, the performance can further boost.

**Effect of the copy-paste strategy.** To analyze the

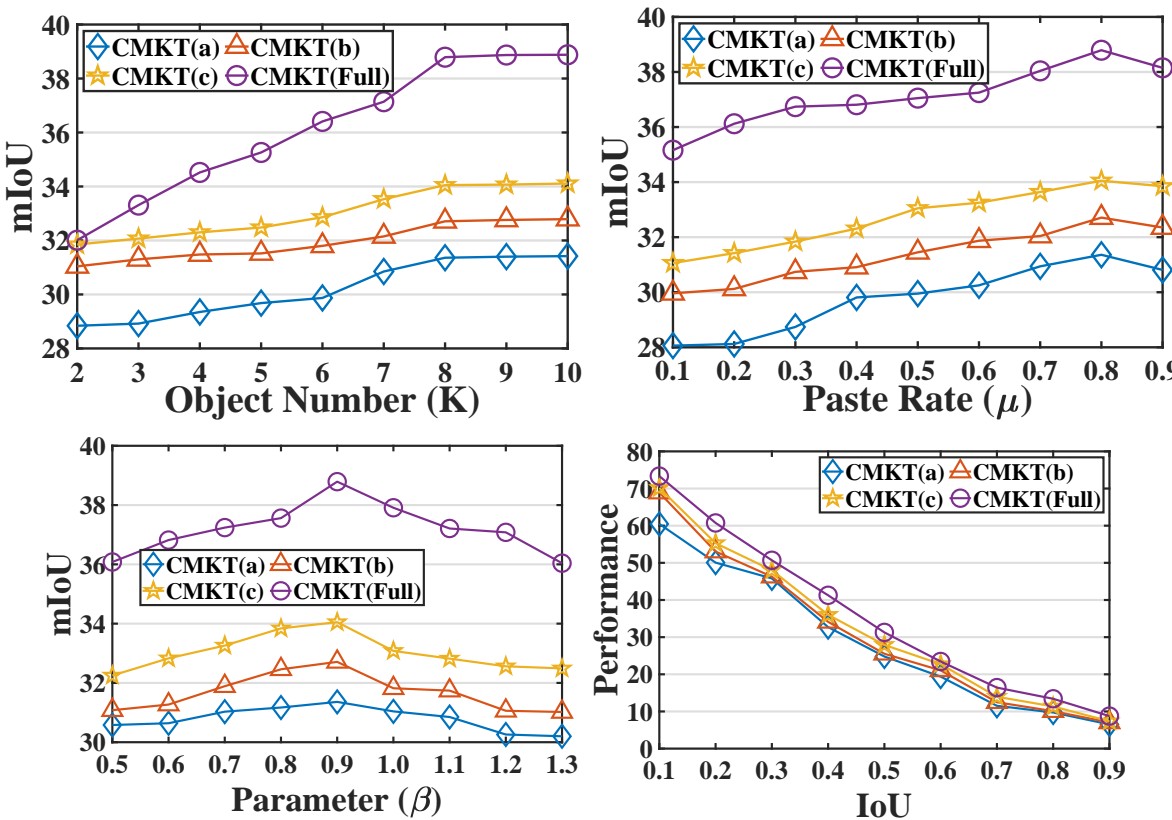

Figure 3: Effect of parameters on ActivityNet Captions.

performance of our copy-paste strategy, an ablation experiment is conducted in Table 6, where "w/ video connection" means that we direct connect two videos for data synthesis. Thus, the copy-paste strategy is more effective to synthesize the multi-action videos for better grounding performance.

**Analysis on the hyper-parameters.** As shown in Figure 3, we investigate the impact of four hyper-parameters: object number ($K$); paste rate ($\mu$); positive parameter ($\beta$); evaluation metrics (IoU). (i) With larger $K$, our proposed CMKT performs better. To balance the performance and the computation cost of object detection, we choose $K = 8$. (ii) All the variants obtain the best performance when the paste rate $\mu$ is set to 0.8. (iii) Our CMKT obtains the best performance when $\beta = 0.9$. (iv) CMKT(Full) obtains the best performance on all IoUs, demonstrating the effectiveness of each module. Thus, we set $K = 8$, $\mu = 0.8$ and $\beta = 0.9$ in all the experiments.

### 4.5 Qualitative Results

As shown in Figure 4, we visualize the localization results. Our CMKT can predict more precise segment boundaries than fully-supervised model 2D-TAN and weakly-supervised method WSTAN.

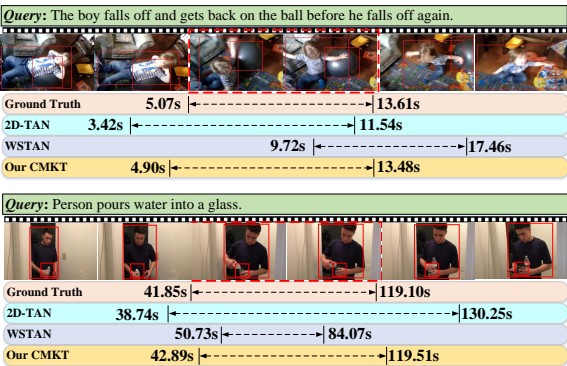

Figure 4: Qualitative results sampled from ActivityNet Captions (top) and Charades-STA (bottom).

## 5 Conclusion

In this paper, we address the challenging yet practical unsupervised TSG task from a brand-new perspective of knowledge transfer. Without further grounding training, we can directly utilize the general knowledge from other cross-modal tasks to guide to match the video and query for retrieving the target segment. To our best knowledge, we make the first attempt to transfer the knowledge from other multi-modal topics to our TSG task. Experimental results validate the effectiveness of CMKT, outperforming the existing unsupervised method by a large margin.

## 6 Acknowledgements

This work is supported by National Natural Science Foundation of China (NSFC) under Grant No. 61972448.

## 7 Limitations

This work analyzes an interesting yet challenging problem of how to transfer the annotation knowledge from other cheap cross-modal annotations to the complicated and challenging unsupervised TSG task. Although we can transfer the knowledge from other cheap multi-modal datasets, the contexts of these datasets are not always helpful for our grounding task. However, our proposed CMKT still provides a brand-new and interesting idea, cross-modal knowledge transfer, for promoting the development of these areas. Therefore, a more contextual and generalizable way to transfer cross-modal knowledge between different topics is a promising future direction.

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
