# OpenReview forum: "Annotations Are Not All You Need: A Cross-modal Knowledge Transfer Network for Unsupervised Temporal Sentence Grounding"
_EMNLP/2023/Conference — EMNLP 2023 Findings_

### Official Review · Reviewer_MGd2 · 2023-08-02

**Typos Grammar Style And Presentation Improvements:** presentations are too complicated.
**Soundness:** 1

**Excitement:**

2: Mediocre: This paper makes marginal contributions (vs non-contemporaneous work), so I would rather not see it in the conference.

**Missing References:**

see weakness

**Paper Topic And Main Contributions:**

This paper performs video moment retrieval under the training of image-noun pairs and video-verb pairs. Transfer learning is applied to incorporate the alignment knowledge.

**Questions For The Authors:**

see weakness

**Reasons To Accept:**

[+] Unsupervised setting for TSG

[+] Sufficient experiments

[+] Enhanced retrieval performances

**Reasons To Reject:**

[-] Although this paper is not the first paper to perform unsupervised TSG, the introduction is unnecessarily written as if it were the first time performing an unsupervised task. There are several citations about TSG and weakly-supervised TSG, but why did the author not cite the previous works about unsupervised TSG in Introduction. See below recent works of unsupervised TSG
- Zero-shot natural language video localization. ICCV'21
- Prompt-based zero-shot video moment retrieval ACM MM'22

[-] This work is more closed to weakly-supervised training, the model trained with a video retrieval dataset and performing retrieval on the video is the process of weakly-supervised setting. Therefore this work has a fatal misunderstanding of contributions.

[-] The figure is too complicated, which ruins the readability.

[-] Many recent works about unsupervised TSG are not cited.
- Zero-shot natural language video localization. ICCV'21
- Prompt-based zero-shot video moment retrieval ACM MM'22

**Reproducibility:**

3: Could reproduce the results with some difficulty. The settings of parameters are underspecified or subjectively determined; the training/evaluation data are not widely available.

**Reviewer Confidence:**

4: Quite sure. I tried to check the important points carefully. It's unlikely, though conceivable, that I missed something that should affect my ratings.

---

> ### Author Rebuttal · Authors · 2023-08-28
>
> **Q1: Introduction is unnecessarily written as if it were the first time performing an unsupervised task. Cite the previous unsupervised TSG works [a,b] in Introduction.**
>
> **[a] Zero-shot natural language video localization. ICCV'21**
>
> **[b] Prompt-based zero-shot video moment retrieval ACM MM'22**
>
> **A1:** Thank you for your patient review. We will reorganize our introduction to avoid misunderstanding.  Also, we will cite and discuss zero-shot TSG methods [a,b] in our revision. Actually, **your mentioned "zero-shot" setting [a,b] is not the same as the "unsupervised" setting.** There is only one paper [c] focusing on the unsupervised TSG. We have already cited and compared it in Table 1, where we achieve better performance. Our "unsupervised" differs from "zero-shot" in two main aspects:
>
> 1) **Annotations are different.** Zero-shot TSG [a,b] means that the input data only contains video set without query set. These works utilize pre-trained object detector and NLP tools to generate language sentence semantically related to the detected segment. Note that, during training, these generated segment-query pairs are semantically aligned. Different from zero-shot setting, the unsupervised TSG means that the input data contains both video and query sets. However, during training, the correspondence between videos and queries are unknown.
>
> 2) **Goal of the designed model.** Since zero-shot methods [a,b] generate semantically aligned segment-query pairs, their models are designed to train with these pseudo labels for learning to ground the query-related segment. This process is the same as the general fully-supervised setting. They severely rely on the quality of the generated sentence.
> Different from them, the unsupervised setting [c] aims to mine the potential relationships between the unknown video and query. It does not need any additional pre-trained model and its model is designed to generalize to the unlabeled cases.
>
> **Overall, the zero-shot setting is quite different from the unsupervised setting.**
>
> [c] Unsupervised temporal video grounding with deep semantic clustering, AAAI’22
>
> **Q2: This work is more closed to weakly-supervised training.**
>
> **A2:** Firstly, we want to clarify that our main contribution is not the unsupervised setting, but is to propose to transfer knowledge from other annotation-cheap tasks into TSG task without using TSG annotations.
>
> Secondly, our work is not the weakly-supervised setting. Generally, the weakly-supervised learning is based on the matched video-query pair without frame-level annotation on the TSG datasets as supervision, while our unsupervised method does not utilize any video-query pair information on the TSG datasets. Both the appearance knowledge and action knowledge are pre-trained from two modules respectively. These two collected knowledges are fixed during our unsupervised framework fine-tuning (i.e., knowledge transfer).
>
> We will highlight our contributions and add more discussion about different settings in our revision.
>
> **Q3: The figure is too complicated.**
>
> **A3:** We will revise our figures to further improve the readability.
>
> **Q4: Recent unsupervised TSG works are not cited.**
>
> **[a] Zero-shot natural language video localization. ICCV'21**
>
> **[b] Prompt-based zero-shot video moment retrieval ACM MM'22**
>
> **A4:** In our revised version, we will provide more analysis of these zero-shot TSG works and provide more details about the difference between zero-shot TSG and unsupervised TSG as:
>
> As the pioneering work of zero-shot TSG, [a] introduces two modules (temporal event proposal (TEP) and pseudo-query (PQ) generation) into a supervised TSG model. At training, pseudo-supervision composed of TEPs and corresponding PQs as a simplified sentence is generated to train the NLVL model. At inference, a natural sentence query is transformed into a simplified one, and temporal segment boundaries are predicted with the trained model.
>
> As a respectable work of zero-shot TSG, [b] design two novel modules (Proposal Prompt and Verb Prompt) into a supervised TSG model. During training, [b] uses pseudo-supervision to train the TSG model. The pseudo-supervision comprises temporal proposals and corresponding pseudo queries (i.e., simplified sentences consisting of nouns and verb prompts). During inference, [b] transforms a sentence query to a simplified one and localizes temporal boundaries with the trained TSG model.
>
> **However, the zero-shot setting is quite different from the unsupervised setting:**
>
> Zero-shot TSG means that the input data only contains video set without query set. These works utilize pre-trained object detector and NLP tools to generate language sentence semantically related to the detected segment. Different from zero-shot setting, the unsupervised TSG means that the input data contains both video and query sets. Moreover, since zero-shot methods generate semantically aligned segment-query pairs, their models are designed to train with these pseudo labels for learning to ground the query-related segment under the fully-supervised setting.
>
> Different from Zero-shot TSG, unsupervised TSG aims to mine the potential relationships between the unknown video and query. It does not need any additional pre-trained model and its model is designed to generalize to the unlabeled cases.

---

### Official Review · Reviewer_4roi · 2023-08-04

**Soundness:** 4

**Excitement:**

4: Strong: This paper deepens the understanding of some phenomenon or lowers the barriers to an existing research direction.

**Missing References:**

[1] Mun J, Cho M, Han B. Local-global video-text interactions for temporal grounding[C]//Proceedings of the IEEE/CVF Conference on Computer Vision and Pattern Recognition. 2020: 10810-10819.

[2] Li M, Wang T, Zhang H, et al. End-to-End Modeling via Information Tree for One-Shot Natural Language Spatial Video Grounding[C]//Proceedings of the 60th Annual Meeting of the Association for Computational Linguistics (Volume 1: Long Papers). 2022: 8707-8717.


**Paper Topic And Main Contributions:**

The paper proposes a novel approach for unsupervised temporal sentence grounding (TSG) without relying on expensive video-query annotations. The proposed Cross Modal Knowledge Transferring (CMKT) network leverages simple cross-modal alignment knowledge from other tasks to model appearance and action information in videos. The extracted knowledge is then fine-tuned and transferred to the TSG task for inference without further training. Extensive experiments on two challenging datasets demonstrate the effectiveness of the proposed approach, outperforming existing unsupervised methods and even competitively beating supervised methods.

**Reasons To Accept:**

1. The proposed approach leverages simple and cheap cross-modal alignment knowledge from other tasks to model appearance and action information in videos, which significantly reduces the reliance on expensive video-query annotations and the need for large-scale training data.

2. The proposed approach introduces a novel copy-paste approach to synthesize various multi-action clips and improve the generalization ability of the model, which enables it to handle complicated videos with multiple action contexts.

3. The proposed approach achieves state-of-the-art performance on two challenging datasets, outperforming existing unsupervised TSG methods and even competing with supervised methods, demonstrating its effectiveness and potential for practical applications.

**Reasons To Reject:**

1. The content of the model diagram is too complicated, and the font is too small, which greatly affects readability.

2. It can be observed that the methodology section occupies a large portion of the paper, containing a significant amount of content with insufficient emphasis on key information. It might be worth considering highlighting the crucial information and moving some of the content to the appendix.

3. The methodology involves a highly complex process, and the model lacks simplicity.

4. There is a lack of organization and analysis of important references in the paper.

**Reproducibility:**

3: Could reproduce the results with some difficulty. The settings of parameters are underspecified or subjectively determined; the training/evaluation data are not widely available.

**Reviewer Confidence:**

3: Pretty sure, but there's a chance I missed something. Although I have a good feel for this area in general, I did not carefully check the paper's details, e.g., the math, experimental design, or novelty.

---

> ### Author Rebuttal · Authors · 2023-08-28
>
> **Q1: Model diagram is too complicated, and the font is too small.**
>
> **A1:** Thank you for your constructive suggestions. We will divide the model diagram into three figures (appearance knowledge collection, action knowledge collection, and knowledge transfer to unsupervised TSG) to improve readability. Also, we will increase the font size and add more explicit explanations in our revision.
>
> **Q2: Highlight crucial information and move some content to appendix.**
>
> **A2:** We will follow your suggestions to highlight the crucial information and move some of the content to the appendix in our revision.
>
> **Q3: Methodology involves complex process, and model lacks simplicity.**
>
> **A3:** We will simply our model for better readability in the future version. Specifically, our model can be summarized as:
>
> (i) We design two knowledge collection modules to extract both appearance and action knowledges from other simple yet cheap cross-modal tasks.
>
> (ii) To fit the simple action knowledge into more complicated videos in TSG, we design an effective copy-paste strategy to synthesize videos with multiple actions.
>
> (iii) To transfer the knowledge into TSG, we obtain parsed words and visual features from the TSG data, and re-represent them by searching and aggregating the consensus features from the two knowledge modules.
>
> (iv) A cycle-consistent loss is also introduced to assist the knowledge transfer.
>
> **Q4: Lack of organization and analysis of some references:**
>
> **[1] Local-global video-text interactions for temporal grounding.**
>
> **[2] End-to-End Modeling via Information Tree for One-Shot Natural Language Spatial Video Grounding.**
>
> **A4:** Actually, we have cited LGI [1] in Table 1 of our paper.  In our revision, we will cite paper [2] and provide more discussion about [1,2]:
>
> [1] is a fully-supervised TSG method, which tackles the TSG task using a regression-based model that learns to extract a collection of mid-level features for semantic phrases in a text query, and to reflect bi-modal interactions between the linguistic features of the query and the visual features of the video in multiple levels.
>
>  [2] is a natural language spatial video grounding method, rather than TSG method. It aims to detect the relevant objects in video frames with descriptive sentences as the query. Especially, it designs a tree-structure approach, and learns to ground natural language in all video frames with solely one frame labeled, in an end-to-end one-shot manner.
>
> Papers [1,2] severely rely on the numerous data annotations. Different from them, our method does not need any TSG annotations. Instead, we target to transfer the annotation knowledge from other cheap cross-modal tasks to the complicated and challenging TSG task. Our unsupervised method achieves very competitive performance compared to existing fully- and weakly-supervised methods.

---

### Official Review · Reviewer_Zmbu · 2023-08-04

**Soundness:** 4

**Excitement:**

4: Strong: This paper deepens the understanding of some phenomenon or lowers the barriers to an existing research direction.

**Paper Topic And Main Contributions:**

This paper addresses the task of unsupervised temporal sentence grounding (TSG). The main contributions of this paper are as follows:
1. This work is the first attempt to use transfer learning knowledge to tackle the unsupervised temporal sentence grounding task. It eliminates the need for any annotations related to TSG data and simplifies the training process.
2. The authors adopted carefully designed techniques to acquire and utilize cross-modal knowledge obtained through transfer learning, aiming to achieve better generalization performance.
3. The model performs well on both datasets, especially in the unsupervised task setting. It also demonstrates competitive performance compared to some models that adopt weakly supervised approaches.

**Reasons To Accept:**

1. There has been limited research on unsupervised temporal sentence grounding in the past. Previous works often focused on meticulously designing models and dealing with complex training processes. However, this work takes a different approach by leveraging transfer learning, which directly eliminates the need for an extensive training process.
2. This article’s approach involves decomposing the original task, enabling the direct transfer of knowledge from other simpler tasks to the unsupervised Temporal Sentence Grounding (TSG) task. During the knowledge acquisition stage, the authors introduced several techniques to enhance the generalization of the acquired knowledge. In the knowledge inference stage, they also addressed the issue of knowledge applicability and proposed specific optimization measures to utilize the knowledge effectively.
3. The experimental section of this work is comprehensive, and the model’s performance shows significant improvements. Moreover, various ablation experiments conducted in the study demonstrate the effectiveness of each proposed design.

**Reasons To Reject:**

1. Although this work eliminates the complex training process for Temporal Sentence Grounding (TSG), there will still be some degree of training cost during the stages of acquiring transfer knowledge and improving knowledge applicability, both in terms of computational resources and time cost. So, compared to the original training process, does it show improvement?
2. The model framework diagram presented in the paper is not sufficiently clear and aesthetically pleasing. There is an excessive use of symbols instead of explicit explanations, which significantly impacts the reading experience.

**Reproducibility:**

4: Could mostly reproduce the results, but there may be some variation because of sample variance or minor variations in their interpretation of the protocol or method.

**Reviewer Confidence:**

4: Quite sure. I tried to check the important points carefully. It's unlikely, though conceivable, that I missed something that should affect my ratings.

---

> ### Author Rebuttal · Authors · 2023-08-28
>
> **Q1: Compare acquiring transfer knowledge and the original training process in terms of computational resources and time cost.**
>
>
> **A1:** Thanks for your valuable comment. We report the time cost of each component of our model and the original training of previous methods for detailed comparison. Since previous unsupervised TSG method (DSCNet) does not provide its code publicly, we only compare our unsupervised CMKT with representative open-source fully- and weakly-supervised methods. We conduct all the experiments on 8 NVIDIA A100 GPUs with a total GPU memory of 640GB. Besides, we measure all the data with strict adherence to the source code in the same environment.
>
> (1) Firstly, we compare our knowledge transfer module with original training as follows:
>
> | Method | mIoU(Act) | Time(Act) | mIoU(Cha) | Time(Cha) |
> |:--:|:--:|:--:|:--:|:--:|
> | CTRL (FS) | 20.54 | 5.31h (original training) | 18.94 | 2.81h (original training) |
> | CNM (WS) | 37.14 | 4.88h (original training) |  **38.11** | 2.34h (original training) |
> | LCNet (WS) | 34.29 | 4.25h (original training) | 38.04 | 2.20 (original training) |
> | WSTAN (WS) | 35.72 | 3.87h (original training) | 27.43 | 1.78h (original training) |
> | **Our CMKT (US)** | **38.79** | **1.14h (knowledge transfer)** | 30.42 | **0.38h (knowledge transfer)** |
>
> Here, "Act" denotes "ActivityNet Captions", "Cha" denotes "Charades-STA", and "Time cost" denotes the time required to complete the training on the whole dataset. Obviously, our knowledge transfer can significantly reduce the time cost of the grounding model with competitive performance. The reason is: As a fully-supervised method, CTRL calculates the relationship between each frame and the given language query, spending much time during training. For weakly-supervised methods (CNM, LCNet and WSTAN), they still spend much time on cross-modal alignment. Different from them, our unsupervised CMKT avoids complex cross-modal alignment computation by our computation-efficient knowledge transfer module, which reduces the time cost.
>
> (2) Secondly, for our knowledge collection modules, their time costs are 10h (appearance knowledge collection) and 43h (action knowledge collection). **Please note that, our knowledge collection modules are the pre-processing modules, which are pre-trained offline. Other downstream methods/tasks can directly utilize our knowledge collection modules to further improve their generalization-ability for avoiding complex training processes.** To assess their effectiveness, we use them as a plug-and-play module for compared supervised methods. Then, we fine-tune their hyper-parameters to obtain the optimal values. We report the plug-and-play study as follows:
>
> | Method | Time(Origin) on Act | Time(Ours) on Act | mIoU(Origin) on Act | mIoU(Ours) on Act | Time(Origin) on Cha| Time(Ours) on Cha| mIoU(Origin) on Cha| mIoU(Ours) on Cha|
> |:--:|:--:|:--:|:--:|:--:|:--:|:--:|:--:|:--:|
> | CTRL (FS) | 5.31h | **4.13h** | 20.54 | **23.32** | 2.81h | **1.93h** | 18.94 | **20.31** |
> | CNM (WS) | 4.88h | **4.06h** | 37.14 | **37.89** | 2.34h | **1.35h** | 38.11 | **38.95** |
> | LCNet (WS) | 4.25h | **3.74h** | 34.29 | **36.01** | 2.20h | **1.28h** | 38.94 | **40.26** |
> | WSTAN (WS) | 3.87h | **3.18h** | 35.72 | **36.24** | 1.78h | **1.04h** | 27.43 | **28.57** |
>
> In the above table, we choose the official codes as the baseline (i.e., "Origin") under the official settings. In "Ours" setting, we first utilize our pretrained two knowledge modules as the pretrained knowledge base. Then, we utilize our knowledge transfer module to enrich the grounding knowledge. Finally, we utilize their remaining modules for video grounding. Obviously, by using our model as the plug-and-play module, other TSG models can significantly reduce the training time and obtain better performance.
>
> Obviously, our CMKT is more efficient than other supervised methods with competitive performance. Besides, our model can serve as a plug-and-play module to inspire some cross-modal downstream tasks for further improving their effectiveness and effectiveness.
>
> **Q2. Model framework diagram is not sufficiently clear. There is an excessive use of symbols instead of explicit explanations.**
>
> **A2.** We will divide the model framework diagram into three figures (appearance knowledge collection, action knowledge collection, and knowledge transfer to unsupervised TSG) to improve the reading experience. Moreover, we will reduce unnecessary symbols and add more explicit explanations for a clearer framework in the revised version.

---

### Meta-Review · Area_Chair_fHDy · 2023-09-18

**Recommendation:** 3

**Metareview:**

This paper received a mixed reviews from 3 reviewers. Reviewers Zmbu and 4roi recommend acceptance given the strong perfromance of the method and the potential of elimate annotation cost. While all reviewers recommend rejection as the paper is overall not well-written and is hard to understand, e.g., complicated figures and excessive use of symbols. Reviewer MGd2 also mentioned the work unnecessarily claimed as first work for unsupervised TSG. AC values the contributions of targeting reducing the use of annotations in the TSG task, however, AC does not agree with the authors that this work should be described as "truly unsupervised" due to its use of annotated assets (Visual Genome and Kinetics). There are work that use purely unannotated data for the task, e.g., "Zero-shot Video Moment Retrieval With Off-the-Shelf Models".

---

### Decision · Program_Chairs · 2023-10-07

**Decision:**

Accept-Findings

**Comment:**

This paper received a mixed reviews from 3 reviewers. Reviewers Zmbu and 4roi recommend acceptance given the strong perfromance of the method and the potential of elimate annotation cost. While all reviewers recommend rejection as the paper is overall not well-written and is hard to understand, e.g., complicated figures and excessive use of symbols. Reviewer MGd2 also mentioned the work unnecessarily claimed as first work for unsupervised TSG. AC values the contributions of targeting reducing the use of annotations in the TSG task, however, AC does not agree with the authors that this work should be described as "truly unsupervised" due to its use of annotated assets (Visual Genome and Kinetics). There are work that use purely unannotated data for the task, e.g., "Zero-shot Video Moment Retrieval With Off-the-Shelf Models".